# Phage Therapy vs. the Use of Antibiotics in the Treatment of *Salmonella*-Infected Chickens: Comparison of Effects on Hematological Parameters and Selected Biochemical Markers

**DOI:** 10.3390/antibiotics11121787

**Published:** 2022-12-09

**Authors:** Łukasz Grabowski, Grzegorz Węgrzyn, Alicja Węgrzyn, Magdalena Podlacha

**Affiliations:** 1Laboratory of Phage Therapy, Institute of Biochemistry and Biophysics, Polish Academy of Sciences, Kładki 24, 80-822 Gdansk, Poland; 2Department of Molecular Biology, Faculty of Biology, University of Gdansk, Wita Stwosza 59, 80-308 Gdansk, Poland

**Keywords:** phage therapy, *Salmonella* infection, chicken, antibiotics, phage cocktail

## Abstract

Previous studies indicated that the use of a phage cocktail, composed of bacteriophages vB_SenM-2 and vB_Sen-TO17, is effective in killing cells of *Salmonella enterica* serovars Typhimurium and Enteritidis in vitro and in the *Galleria mellonella* animal model as efficiently as antibiotics (enrofloxacin or colistin) and induced fewer deleterious changes in immune responses. Here, we investigated the effects of this phage cocktail on the hematological parameters and selected biochemical markers in chickens infected with *S. enterica* serovar Typhimurium, in comparison to those caused by enrofloxacin or colistin. We found that treatment with antibiotics (especially with enrofloxacin) caused nonbeneficial effects on red blood cell parameters, including hematocrit, MCV, MCH, and MCHC. However, *Salmonella*-induced changes in the aforementioned parameters were normalized by the use of the phage cocktail. Importantly, hepatotoxicity was suggested to be induced by both antibiotics on the basis of increased alanine transaminase (ALT) and aspartate aminotransferase (AST) activities, in contrast to the phage cocktail, which did not influence these enzymes. We conclude that phage therapy with the cocktail of vB_SenM-2 and vB_Sen-TO17 in *Salmonella*-infected chickens is not only as effective as antibiotics but also significantly safer for the birds than enrofloxacin and colistin.

## 1. Introduction

Salmonellosis is one of the most serious forms of infectious disease affecting poultry. Although *Salmonella enterica* is not the most dangerous pathogen to birds, it can lead to serious food-borne disease in humans [1,2,3]. Almost 100 million cases of salmonellosis are detected every year in humans; among which, more than 150 thousand are fatal cases. Therefore, when poultry infection with *Salmonella* is detected, the meat is disqualified from the market [4]. The appearance of *S. enterica* strains that are resistant to many antibiotics makes the problem even more serious [1,5,6,7]. In order to prevent or limit the development of antibiotic resistance, the use of antibiotics in livestock feed has been forbidden in the European Union, and there are partial restrictions in the United States [4,8,9]. For this reason, the development of novel therapies and approaches to protect poultry from *S. enterica* infection is required [5,10,11,12,13].

Our research group has previously contributed to the development of novel therapies and/or preventive treatments against *S. enterica* by creating a cocktail of the bacteriophages vB_SenM-2 and vB_Sen-TO17, which infect serovars Typhimurium and Enteritidis [14]. The cocktail was characterized in vitro to be effective at killing *S. enterica* cells and reducing the bacterial biofilm. Moreover, safety was initially assessed by the limulus (LAL) test and by determining the viability of chicken fibroblasts. The treatment of *S. enterica*-infected *Galleria mellonella* larvae with the cocktail improved their survival rate [14]. These results provided a basis to perform additional studies using experimental infection on chickens with *S. enterica* serovar Typhimurium. Overall, phage therapy was found to be as effective as therapy with either enrofloxacin or colistin and appeared less deleterious, inducing fewer changes in the microbiome than antibiotics [15]. This conclusion was corroborated in subsequent analyses in which the levels of several pro- and anti-inflammatory cytokines were measured, and the relative levels of the stress hormones were determined. The analyses showed, for the first time, that the phage cocktail did not disturb the immune homeostasis in chickens, while treatment with antibiotics (enrofloxacin or colistin) caused cytokine imbalance, changes in proportions between immune cell subpopulations, and stress axis hyperactivity [16]. Thus, antibiotics (especially enrofloxacin) may cause serious adverse effects in poultry, while phage therapy appears safer [17].

To further assess the safety of phage therapy vs. antibiotics (enrofloxacin and colistin) on *S. enterica*-infected chickens, we measured the hematological parameters and levels of selected biochemical markers in blood samples withdrawn from tested birds. The main goal of these experiments was to further determine, compare, and detect any potential adverse effects of these two kinds of antibacterial therapies. Such studies should facilitate the identification of the most efficacious and safe methods for the treatment of infected poultry.

## 2. Results

### 2.1. Changes in the Counts of Blood Morphotic Elements of Chickens Receiving Phage Cocktail or Antibiotics

Whether bacteriophages may affect the number of blood-morphotic elements has been examined; therefore, a comprehensive analysis of this phenomenon was completed. The absolute monocyte count, absolute eosinophiles count, erythrocyte count, hemoglobin level, hematocrit, mean corpuscular volume (MCV), mean corpuscular hemoglobin (MCH), mean corpuscular hemoglobin concentration (MCHC), and platelet count were examined.

The chickens treated with enrofloxacin (group 4) had significantly decreased erythrocyte counts (Figure 1) and hematocrit (Figure 2) throughout the experiment compared to the uninfected control groups. The increased erythrocyte counts in group 3, as compared to groups 1 and 2, were evident throughout the experiment, especially in comparison with group 1. It was observed that the red blood cell count in the phage-treated group one day after infection (group 6) was slightly decreased when compared to the phage-treated control group. However, this value normalized at termination 2, termination 3, and termination 4.

Regarding the platelet count (Figure 3), a slight decrease was observed in the infected control group and the colistin-treated group compared to the saline-treated control group, but these values normalized later (termination 2 and termination 3).

The values of the absolute monocyte count, absolute eosinophilia count, mean corpuscular volume (MCV), mean corpuscular hemoglobin (MCH), and mean corpuscular hemoglobin concentration (MCHC) are shown in Table 1. In the case of group 3, a drastic increase in the total monocyte count and the eosinophil count was noted during the experiment. Interestingly, the values of these two parameters in the groups treated with antibiotics (group 4 and group 5) were drastically reduced as compared to the uninfected control groups throughout the experiment. On the other hand, in animals that were treated with a phage cocktail one day after infection (group 6), as well as two days after the detection of bacteria in the feces (group 7), no statistically significant changes were observed relative to the uninfected phage groups. However, it was noted that, in the case of the group treated with the phage cocktail (group 8), the final counts of monocytes and eosinophils were significantly elevated relative to the uninfected control groups, while they were significantly lower than the values obtained in the infected control group (group 3). Moreover, in the groups treated with antibiotics (group 4 and group 5), the Mean Corpuscular Volume (MCV) values were initially (termination 1 and termination 2) not significantly different from the uninfected control groups, but at further stages of the experiment, they started to decrease and became statistically significantly different from the uninfected control groups. In the case of groups 6 and 7, the MCV levels were not significantly different from the uninfected control groups, contrary to group 8, which, from termination 2 to the end of the experiment, significantly increased in comparison to the control group receiving saline.

The Mean Corpuscular Hemoglobin (MCH) values were significantly decreased throughout the experiment in the infected control group. A drastic increase was observed in the enrofloxacin-treated group compared to the saline-treated control group at termination 1, termination 3, and termination 4. Interestingly, in colistin-treated animals, despite the initial lack of difference from the uninfected control groups, an increase was observed compared to the saline-treated control group (group 1) at the end of the experiment (termination 4). Although no significant differences were observed in group 6 relative to the uninfected control groups, such differences were observed in groups 7 and 8 at termination 2 and termination 4. Additionally, it was observed that the Mean Corpuscular Hemoglobin Concentration (MCHC) values in group 8, although not significantly different from the uninfected control groups at the beginning of the experiment, significantly decreased at termination 3 and termination 4. No significant changes were observed in group 6 and group 7 compared to the uninfected control groups throughout the experiment. Interestingly, the MCHC in the colistin-treated group, although initially significantly increased at termination 1 and termination 2 relative to the uninfected control groups, normalized at the end of the experiment (termination 4) and was not significantly different from the uninfected control groups. The enrofloxacin-treated group noted elevated MCHC as compared to the uninfected control groups at termination 1, termination 2, and termination 4, contrary to the infected control group, whose values were significantly decreased.

### 2.2. Blood Biochemical Parameters in Plasma of Chickens Subjected to Phage Therapy and Antibiotic Therapy

The levels of alanine transaminase (ALT), aspartate aminotransferase (AST), and c-reactive protein (CRP) in the blood plasma of chickens that were treated with phage therapy or antibiotic therapy were investigated.

The results showed a significant increase in the ALT levels in group 3, which were infected with *Salmonella* but not treated throughout the experiment (Figure 4). Interestingly, an even greater deviation from the reference value manifested in the significantly elevated level of the enzyme, which indicates liver dysfunction or even damage, was observed in group 5, who were receiving colistin. Furthermore, a hepatotoxic effect was also evident in group 4, in which enrofloxacin was administered. In contrast, when the phage cocktail was used, as well as in the *Salmonella*-infected groups where it was applied 24 h after infection (group 6) or after 2 days (group 7), the level of the tested enzyme remained low, within the normal range throughout the experiment. Only in group 8, in which phage cocktail administration was implemented 4 days after the development of bacterial infection, the level of the examined parameter was significantly increased. An analogous direction of change was also observed for the second indicator of liver dysfunction, AST (Figure 5).

With regard to CRP (Figure 6), its significantly elevated level was shown only in *Salmonella*-infected group 3 and group 8, where the phage cocktail was implemented later. The increase in CRP release by the liver into the peripheral blood is an indicator of the development of severe inflammation in the organism.

## 3. Discussion

Phage therapy is considered to be a possible alternative to the use of antibiotics to combat pathogenic bacteria [18]. Although the introduction of this method as an officially approved therapy to treat patients requires both intensive studies and changes in the current law, it appears that the use of bacteriophages in the treatment of infected animals might be possible significantly earlier, especially due to differences in regulations between medical and veterinary rules [19]. Nevertheless, before formal recommendations to use phage therapy to treat animals can be issued, both the efficacy and safety of this method should be confirmed.

Our previous studies led to the development of a phage cocktail, composed of bacteriophages vB_SenM-2 and vB_Sen-TO17, which is effective in eliminating *S. enterica* serovars Typhimurium and Enteritidis and safe to the *G. mellonella* animal model, as well as in cell culture tests [14]. Moreover, this cocktail was as effective in the treatment of *S. enterica*-infected chickens as antibiotics (enrofloxacin or colistin) while causing significantly less adverse effects than those drugs, as estimated by changes in the gut microbiome [15] or disturbances in the immune response [16]. On the other hand, the effects of these two methods of treatment were not previously compared in light of changes in the hematological parameters and biochemical markers. Thus, this work was conducted to fill this gap in our knowledge.

Hematocrit is an indicator that determines the volume ratio of erythrocytes to whole blood. A decrease in this index, as well as an insufficient erythrocyte count, can indicate a number of abnormalities, such as gastrointestinal bleeding, bone marrow disturbances, kidney damage, or, most commonly, the onset of anemia resulting from iron deficiency. Iron determines the supply of adequate amounts of oxygen, electron transport, and the proper functioning of enzymes. It is important for high-metabolic rate cells [17]. Hematocrit is of particular importance in the assessment of animal physiology. Some mammals, such as dogs or horses, can store erythrocytes in the spleen and thus modulate the hematocrit or hemoglobin levels, depending on the intensity of the exercise. Birds, on the other hand, which do not have this regulatory mechanism, can lower their hematocrit values by hemodilution in response to intensive exercise or exposure to aversive factors [18]. When considering the effect of the administered substances on erythrocyte indices, including the hematocrit value, physiological factors such as age, gender and hormone levels, should also be taken into account [19]. Other indicators also play an important role when it comes to assessing the organism’s overall condition. The MCV provides information on the average volume of a single erythrocyte and allows for the early detection of anemia. Its value depends on plasma osmolarity and the rate of erythrocyte division. The MCH is an indicator of the average mass of hemoglobin in a single red blood cell. Together with the MCHC value, it is useful in distinguishing between different types of anemia [19]. Changes in hematological parameters are useful for assessing the organism’s adaptation to adverse conditions or stressors. This is particularly important in the case of industrial poultry-rearing for meat and eggs. Erythrocyte indices change significantly depending on the husbandry conditions and nutritional status. Overly nutrient-poor feed and prolonged exposure of poultry to stress factors result in lower erythrocyte counts and hemoglobin levels, leading to erythrocytopenia and reduced organism performance. An increase in ambient temperature results in the loss of a large amount of liquid through the respiratory system, which then leads to a decrease in the plasma volume and an increase in the hematocrit level. Similarly, in the case of dehydration by the evaporation process, there is a significant increase in the hematocrit values. In contrast, a non-physiological reduction in the hematocrit levels through the hemodilution process occurs, with severe stressors, especially heat stress exposure [20]. As indicated by the results shown in this report, antibiotic therapy, which, in poultry, is sometimes administered prophylactically for chicken growth promotion or therapeutically [21] can be a type of negative stressor that causes a number of disturbances in the organism, especially if we consider the immune system [16] and, in particular, erythrocyte indices. This is important, because it has a number of negative consequences, including economic ones, due to deteriorations in the quality of the meat, the lower number of eggs laid, or the welfare of consumers (adverse effects on the intestinal microbiome and the development of drug-resistant bacteria) [21]. The search for alternative methods to control bacterial infections, therefore, needs to be addressed in depth, especially regarding infections with *Salmonella* strains, which pose serious challenges to the immune system of animals destined for consumption [22]. It was demonstrated that *Salmonella enterica* serotype Enteritidis leads, in chickens, to the slowed growth of key immune organs, changes in the profile of important immune cell types, reduced antibody production, increased levels of pro-inflammatory cytokines, and excessive activation of the stress axis, with consequent effects on the hematological parameters [23]. This was also confirmed by our observations, which showed, among other things, a non-physiological increase in the erythrocyte count, hematocrit level, and MCV value. This type of deviation from reference values is most often observed in situations of severe dehydration, which result in blood thickening. Although the literature contains significant reports on the negative effects of antibiotic therapy on hematological parameters in poultry, there are papers indicating a number of different side effects of the administration of popular veterinary antibiotics, such as enrofloxacin [24]. Only a few previously published reports addressed the problem investigated in this work. Nevertheless, it appears that the results of all these studies are quite similar [25,26]. A comparison of the efficacy of the phages and antibiotics against acute pneumonia in a mouse model was previously described [27]. In that work, two phages (536_P1 and LM33_P1) and three antibiotics (ceftriaxone, cefoxitin, and imipenem–cilastatin) were used. The phages significantly and rapidly reduced the number of bacterial cells and restored the normal blood counts, which were otherwise disrupted by the development of an abrupt bacterial infection [28]. It should also be emphasized that the rapid lysis of bacterial cells does not induce an increase in inflammatory markers as compared to antibiotic therapy [29]. Our studies indicated that there was no significant change in the number of immunocompetent cells (lymphocytes, monocytes, or neutrophils) in the serum after the administration of bacteriophages vB_SenM-2 and vB_Sen-TO17. Analogous results were previously reported by others regarding changes in the number of macrophages, T and B lymphocytes, and dendritic cells in peripheral organs (spleen, liver, and lymph nodes) after the administration of other phages [30,31,32]. Our results also confirmed that, after the administration of a phage cocktail, the analyzed hematological parameters were not significantly different from the results obtained in the control groups. The normalization of the erythrocyte counts following the phage cocktail had tangible physiological benefits, as it avoided disturbances in the hematological parameters induced by bacterial infection, which led to changes and had a negative impact on many organs, particularly the liver and spleen. However, it should be emphasized that only administration immediately or up to two days after detection of the presence of bacteria in the feces guarantees the effective action of the phage cocktail. When it is administered after a longer period, the bacterial infection is already developed enough to cause a number of negative changes that disrupt the functioning of many systems and organs, including, but not limited to, the spleen and liver, resulting in hepatosplenomegaly [33].

In summary, our results confirmed a favorable safety profile for the use of phage therapy (particularly, the phage cocktail consisting of vB_SenM-2 and vB_Sen-TO17) in *Salmonella*-infected chickens. Moreover, this study indicated serious adverse effects of enrofloxacin and colistin on the hematological parameters and ALT and AST activities in these birds. Therefore, phage therapy with bacteriophages vB_SenM-2 and vB_Sen-TO17 may be further considered as an alternative method to either treat or prevent chicken infections with *S. enterica* serovar Typhimurium.

### Conclusions

Our results indicate significantly fewer adverse effects as a result of the phage cocktail relative to the tested antibiotics. The latter agents caused deleterious changes in the red blood cells parameters, including hematocrit, MCV, MCH, and MCHC. However, *Salmonella*-induced changes in the aforementioned parameters were normalized by the use of the phage cocktail. Furthermore, bacteriophages administered either immediately or two days after infection did not significantly affect the number of lymphocytes, monocytes, and neutrophils in the serum. Administration of the tested antibiotics also caused increased activities of ALT and AST, suggesting the hepatotoxicity of these compounds. This was in contrast to the phage cocktail, which did not influence the activities of these enzymes.

## 4. Materials and Methods

### 4.1. Animals

Details of the animals and experimental conditions were described previously [15,16]. In brief, the experiment was conducted on nongenetically modified chickens (*Gallus gallus domesticus*) that were purchased from a licensed breeder (registration number PL28036602, Poland). The Experimental Infection Pavilon at the Department of Bird Diseases (Faculty of Veterinary Medicine, University of Warmia and Mazury, Olsztyn, Poland) was divided into 8 m^2^ boxes that held 25 chickens each. Living conditions were strictly controlled and monitored, with an average humidity of 75% under conditions of regular light–dark cycles (12 h day/12 h night, at light intensity 10 lx) and forced ventilation with 17 air changes per hour). Temperature was reduced from 33 °C (beginning of the experiment) to 22 °C (the end of the experiment). The chickens had unlimited access to forage and water. To reduce the risk of contamination, the entire research complex was equipped with a high-efficiency particulate-absorbing (HEPA) filter system and automation to maintain a cascade of pressures in the sanitary corridors, boxes, and locks. All the experiments were approved by the Local Ethics Committee for Experimental Animals in Olsztyn (permission no. 62/2019, dated 30 July, 2019).

### 4.2. Bacteriophages and Bacterial Strain

The bacteriophages vB_Sen-TO17 and vB_SenM-2, used for the phage cocktail, were characterized previously [34,35], and their safety was confirmed using in vitro experiments with the chicken fibroblast model (UMNSAH/DF-1) and in vivo studies with the *Galleria mellonella* animal model [14]. *Salmonella enterica* serovar Typhimurium (strain KOS 13) was obtained from the National *Salmonella* Center at the Medical University of Gdansk (Poland), and *Salmonella enterica* serovar Heidelberg came from the Collection of the Department of Molecular Biology, University of Gdansk. Isolation of *S*. Typhimurium in chicken fecal samples and cloacal swabs was conducted in accordance with ISO 65791:2017 standards and the previously described procedure [16].

### 4.3. The Preparation of Phage Pocktail

The preparation of the phage lysates included in the cocktail used in the experiment was carried out in accordance with the previously published protocols [15]. In brief, an overnight culture of *S. enterica* (*S.* Heidelberg for phage vB_SenM-2 and *S*. Typhimurium for vB_SenTO17) after inoculation into fresh LB medium (BioShop, Burlington, ON, Canada) at the ratio (*v*/*v*) of 1:100 and incubation at 37 °C with shaking (150 rpm) until OD_600_ = 0.15 (1.5 × 10^8^ CFU/mL) was infected with the appropriate bacteriophage at a multiplicity of infection (m.o.i) of 0.5 and incubated at 37 °C until lysis. To purify the phage lysate, polyethylene glycol 8000 (PEG8000) (BioShop, Burlington, ON, Canada) was added to a final concentration of 10%, and the lysate was then incubated with shaking overnight at 4 °C, using a magnetic stirrer (Carl Roth, Karlsruhe, Germany). Then, the lysate was centrifuged at 10,000× *g* for 30 min at 4 °C (Avanti JXN-26, rotor JLA-8.100, Beckman Coulter, Indianapolis, IN, USA), and the obtained precipitate was suspended in 0.89% NaCl (Alchem, Torun, Poland). To remove PEG8000 completely, 2 mL of chloroform was added to the lysate, which was subjected to double-centrifugation at 4000× *g* for 15 min at 4 °C (Avanti JXN-26, rotor JS-13.1, Beckman Coulter, Indianapolis, IN, USA). In the next step, the lysate was subjected to ultracentrifugation in a sucrose gradient (Sigma Aldrich, St. Louis, MO, USA) at 95,000× *g* (Optima XPN-100, rotor SW32.1 Ti, Beckman Coulter, Indianapolis, IN, USA) for 2.5 h at 10 °C. To remove residual sucrose, the lysate was dialyzed against 0.89% NaCl for 7 days at 4 °C. In order to exclude the possibility of contamination with bacterial endotoxin, the Purified Thermo Scientific^TM^ LAL Chromogenic Endotoxin Quantitation Kit (Catalog No.: 12117850, Thermo Fisher Scientific Inc., Paisley, UK) was used. The obtained lysates were used to prepare a phage cocktail, which was administered to the chickens. The purified and checked lysates of bacteriophages vB_SenM-2 and vB_Sen-TO17 were mixed in a 1:1 ratio (1 × 10^9^ PFU/mL of each phage). Finally, the cocktail was suspended in 20 mM of CaCO_3_.

### 4.4. Animal Groups and the Schedule of the Treatment

The detailed course of the experiment was described in two previously published papers [15,16]. In brief, two hundred seven-day-old chickens were randomly divided into eight experimental groups: group 1, receiving saline, and group 2, receiving a phage cocktail from day 1 to day 15; these were the controls and were not infected with bacteria. For the former group, the aim was to see if the administration procedure could have a significant effect on the studied parameters, while the latter group was used to test the potential impact of the phage cocktail. Group 3 was the positive control; there were 25 *Salmonella*-infected chickens receiving saline until day 15 of the experiment. At day 0 of the experiment, groups 3–8 were infected by administering 1 mL of *S*. *Typhimurium* (10^6^ CFU/mL) suspended in 0.89% NaCl into the beak. Twenty-four hours after infection (day 1 of the experiment), the chickens in group 4 started receiving enrofloxacin (Scanflox, Scanvet, Warsaw, Poland; dose 10 mg/kg per day), while 25 animals in group 5 were given colistin (Colisol, Ceva Animal Health, Warsaw, Poland; dose 120.000 IU/kg per day). For both groups, administration was continued for 5 days. Groups 6, 7, and 8 received the phage cocktail for 14 days. In the case of group 6, administration began analogously to the antibiotics 24 h after infection, while the animals in groups 7 and 8 started receiving the phage cocktail two and four days after confirmation of the bacteria in feces, respectively. The blood samples were taken at four timepoints, while some of the animals from each group were sacrificed in a CO_2_ chamber. At day 6 of the experiment, after the end of the antibiotic treatment, 5 mL of blood was collected from five chickens of each group (termination 1). At day 20 of the experiment, following the completion of phage therapy in group 6, blood was collected from another 5 chickens (in groups 1–6) and 10 chickens (in groups 7 and 8; termination 2). Subsequent blood sampling and sacrifice were performed at day 28 of the experiment (5 chickens from each group; termination 3) and day 34 of the experiment (10 chickens from each group; termination 4).

### 4.5. Blood Collection

The blood sampling methodology was consistent with the previously published protocols [16]. In brief, blood samples of 5 mL were collected from each chicken. To prevent blood clotting, the heparinized syringes tipped with a 25-gauge, 1-in-long needle, and tubes containing sodium heparin were used. During the procedure, the animals were gently immobilized by holding, and the needle was inserted into the brachial wing vein at a shallow angle (approximately 10–20°). Each blood sample was immediately divided according to the course of further determination: 1 mL of whole blood was used for morphological analyses (monocytes, eosinophils, and red blood cell parameters) and flow cytometry (results of flow cytometric analyses were already published by Grabowski et al. [16]), while the remaining blood was centrifuged (1800× *g* for 15 min at 4 °C) to obtain plasma, which was subjected to deep freezing (−80 °C) until further analysis.

### 4.6. Analysis of Selected Blood Morphological Parameters

The morphological analysis of the collected whole blood sample (200 µL) was performed in the Horiba ABX Micros ES 60 automatic analyzer (Horiba Medical. Japan). Linearity specifications were determined by analyzing dilutions of a commercially available linearity control material that contains no interfering substances. In order to avoid meaningless results due to incorrect counts, the linearity range used for these particular parameters was: 0–99.9 K/µL for WBC; 0–8 M/µL for RBC; 0–24 g/dL for HGB; 50–200 fL for MCV; 0–2000 K/µL for PLT; 5–18 fL for MPV; and 0–30% for RETIC%. The automatic analyzer used is commonly applied for the evaluation of hematological parameters in various animal species [36,37,38]. The following parameters were monitored: absolute number of monocytes and eosinophils, as well as red blood cell system indexes: erythrocyte count, hematocrit (HCT) level, mean red cell volume (MCV), mean corpuscular hemoglobin concentration (MCHC), and platelet number.

### 4.7. Determination of Alanine Transaminase (ALT) and Aspartate Aminotransferase (AST) Concentrations in Peripheral Blood Plasma

Levels of the blood biochemical parameters, such as alanine transaminase (ALT) and aspartate aminotransferase (AST), were determined using an automated Architect c8000 Abbott biochemical analyzer (Abbott, Chicago, IL, USA). This is a fully automated system that performs sample processing using potentiometric and photometric methods. The relevant calibration parameters and chicken-dedicated reference values were configured into the system and validated prior to the main analysis. For the determination of the above parameters, the following reference interval was used: 19–21 (IU/L) for ALT and 131–486 (IU/L) for AST, respectively. This was defined on the basis of the Merck Veterinary Manual (2011).

### 4.8. Determination of C-Reactive Protein (CRP) Concentration (Using ELISA) in Peripheral Blood Plasma

The measurement of the CRP level was carried out using an ELISA immunoenzymatic assay, based on the formation of bonds between antigen and antibody, which are revealed by the color reaction with immunoglobulin-conjugated enzymes and their respective substrates. The procedure was performed according to the manual included in the set of commercially available reagents (Catalog No.: ELK2038, ELK Biotechnology Co., Ltd., Wuhan, China) and the previously described method [16]. All reagents and samples were brought to room temperature (20–25 °C) before use. A total of 100 uL of buffer, test samples, or standards were added in duplicate to each well of the titration plate (96-well Nunc plate) coated with CRP-specific monoclonal antibodies. The plate was covered and incubated at 37 °C for 80 min. Then, the plate contents were drained and washed with the prepared buffer three times to remove an excess of unbound antigens. Next, 100 µL of a solution of specific polyclonal biotinylated antibody, conjugated with the enzyme for CRP, were added. The plate was covered, incubated at 37 °C for 50 min, then drained again and washed three times. Subsequently, 100 µL of streptavidin-labeled horseradish peroxidase enzyme solution was added and incubated for 50 min at 37 °C. The plate was then drained and washed five times, and 90 µL of 3,3′5,5′-tetramethylbenzidine (a colored substrate for horseradish peroxidase) solution was added and incubated for 20 min in the dark at 37 °C. The reaction was stopped by adding 50 µL of the blocking solution, which changed the color of the product (from blue to yellow). Absorbance was measured at 10 min after stopping the reaction using a Multiskan FC microplate reader coupled with Skanlt 6.1.1. RE software (Thermo Fisher Scientific, Waltham, MA, USA), which analyzes the spectrophotometric color intensity, plots a standard curve based on the standards used, and reads the concentration values of the CRP in the tested plasma samples. The results were given in ng/mL. The minimum sensitivity for the test was 0.32–20 ng/mL.

### 4.9. Statistical Analysis

The results are presented as mean ± standard deviation (SD). For statistical analysis of the results, SPSS 21.0 (SPSS Inc., Armonk, NY, USA) software was used. The normality of the distribution of the variables was checked with the Kolmogorov–Smirnov test and the homogeneity of the variances with Levene’s test. If the assumptions of normality of distribution and/or homogeneity of variance were not met, the Kruskal–Wallis test and post hoc Dunn’s test were applied. Once both assumptions were met, the analysis was carried out on the basis of ANOVA and post hoc Tukey’s test. Statistically significant differences were considered when *p* < 0.05.

## Figures and Tables

**Figure 1 antibiotics-11-01787-f001:**
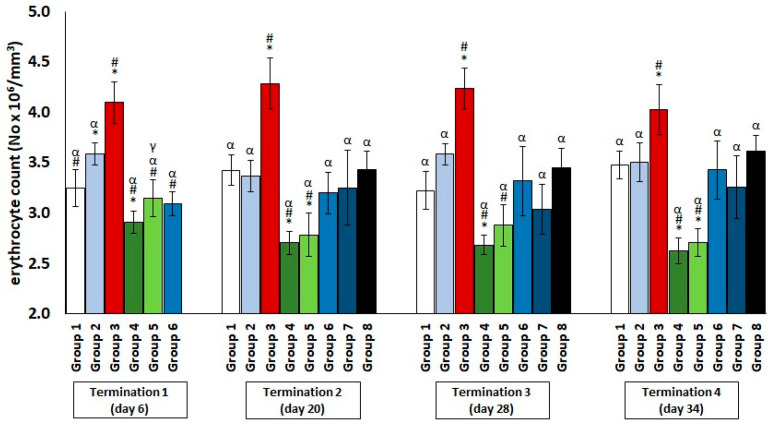
Changes in the erythrocyte count number in the blood of chickens receiving phage cocktail or antibiotics after 6, 20, 28, and 34 days of experiments. Results are presented as mean values ± SD. Statistical analyses were performed by ANOVA and post hoc Tukey’s test. The significance of differences between controls and particular treated groups are observed and marked by asterisks (*) vs. saline control; (#) vs. phage control (group 2); (α) vs. infected control (group 3); (γ) vs. termination 2.

**Figure 2 antibiotics-11-01787-f002:**
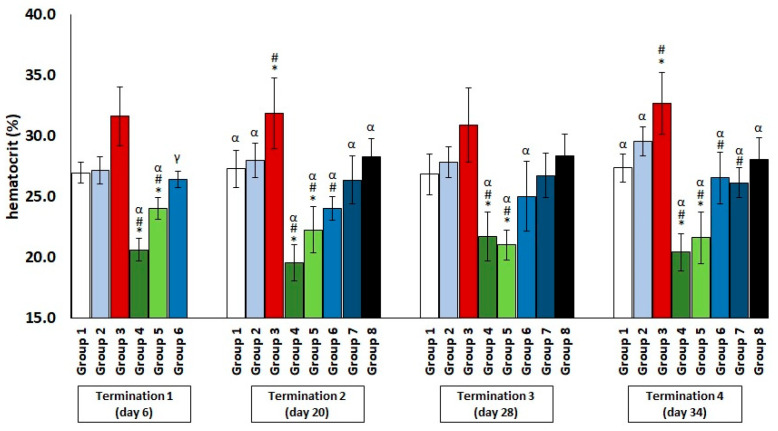
Changes in the hematocrit level in the blood of chickens receiving phage cocktail or antibiotics after 6, 20, 28, and 34 days of experiments. Results are presented as mean values ± SD. Statistical analyses were performed by the Kruskal–Wallis test and post hoc Dunn’s test. The significance of differences between controls and particular treated groups are observed and marked as follows: asterisks (*) vs. saline control; (#) vs. phage control (group 2); (α) vs. infected control (group 3); (γ) vs. termination 2.

**Figure 3 antibiotics-11-01787-f003:**
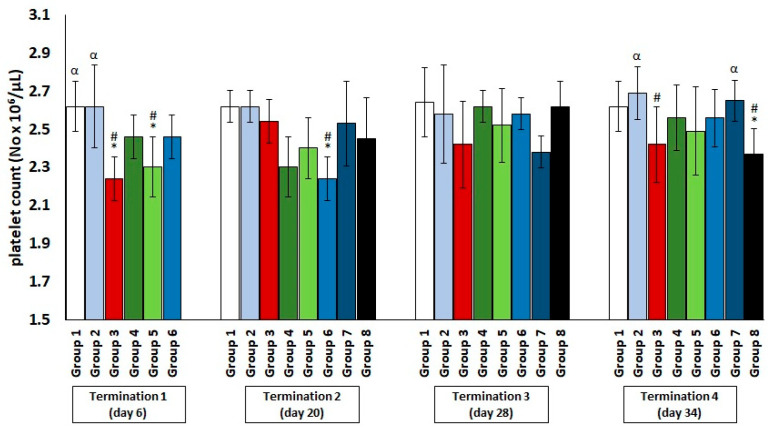
Changes in the platelets count number in the blood of chickens receiving phage cocktail or antibiotics after 6, 20, 28, and 34 days of experiments. Results are presented as mean values ± SD. Statistical analyses were performed by ANOVA and post hoc Tukey’s test. The significance of differences between controls and particular treated groups are observed and marked by asterisks (*) vs. saline control; (#) vs. phage control (group 2); (α) vs. infected control (group 3).

**Figure 4 antibiotics-11-01787-f004:**
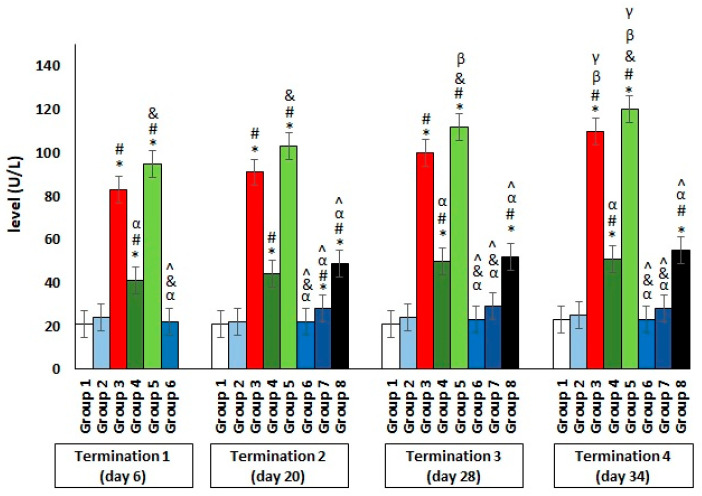
Changes in the alanine transaminase (ALT) level in the blood of chickens receiving the phage cocktail or antibiotics after 6, 20, 28, and 34 days of experiments. Results are presented as mean values ± SD. Statistical analyses were performed by the Kruskal–Wallis test and post hoc Dunn’s test. The significance of the differences between controls and particular treated groups are observed and marked by asterisks (*) vs. saline control; (#) vs. phage control (group 2); (α) vs. infected control (group 3); (&) vs. group 4; (^) vs. group 5; (β) vs. termination 1.

**Figure 5 antibiotics-11-01787-f005:**
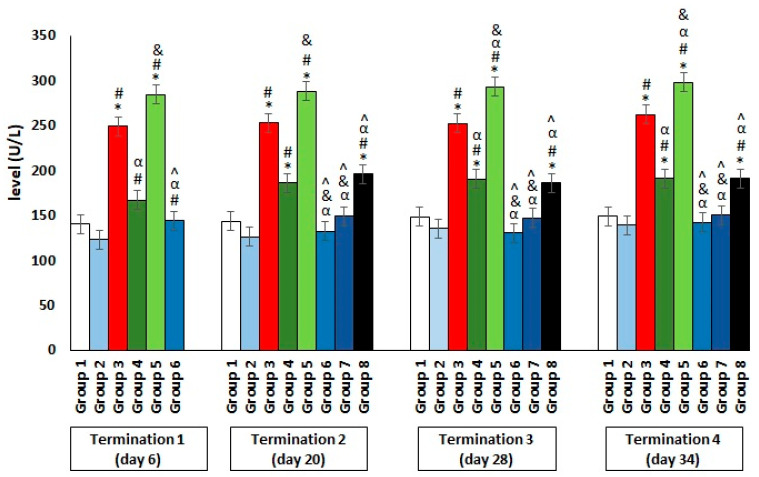
Changes in the aspartate aminotransferase (AST) level in the blood of chickens receiving the phage cocktail or antibiotics after 6, 20, 28, and 34 days of experiments. Results are presented as mean values ± SD. Statistical analyses were performed by the Kruskal–Wallis test and post hoc Dunn’s test. The significance of the differences between controls and particular treated groups are observed and marked by asterisks (*) vs. saline control; (#) vs. phage control (group 2); (α) vs. infected control (group 3); (&) vs. group 4; (^) vs. group 5.

**Figure 6 antibiotics-11-01787-f006:**
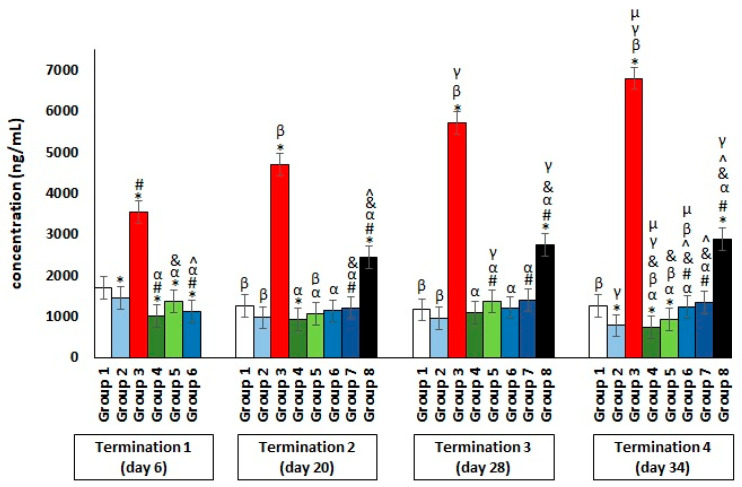
Changes in the C-reactive protein (CRP) concentration in the blood of chickens receiving the phage cocktail or antibiotics after 6, 20, 28, and 34 days of experiments. Results are presented as mean values ± SDs. Statistical analyses were performed by the Kruskal–Wallis test and post hoc Dunn’s test. The significance of differences between controls and particular treated groups are observed and marked by asterisks (*) vs. saline control; (#) vs. phage control (group 2); (α) vs. infected control (group 3); (&) vs. group 4; (^) vs. group 5; (β) vs. termination 1; (γ) vs. termination 2; (µ) vs. termination 3.

**Table 1 antibiotics-11-01787-t001:** Changes in the absolute monocyte count, absolute eosinophil count, Mean Corpuscular Volume (MCV), Mean Corpuscular Hemoglobin (MCH), and Mean Corpuscular Hemoglobin Concentration (MCHC) of chickens receiving the phage cocktail or antibiotics after 6, 20, 28, and 34 days of experiments.

		Absolute Monocyte Count		Absolute Eosinophil Count		MCV		MCH		MCHC	
Termination (day)	Group	(No × 10^9^/L)	*p*	Absolute Eosinophil Count (No × 10^9^/L)	*p*	(fL)	*p*	(pg)	*p*	(g/L)	*p*
Termination 1 (day 6)	Group 1	645.0 ± 84.1	α	449.0 ± 41.9	α	140.8 ± 2.3	α	51.5 ± 1.3	α	354.2 ± 2.0	α
	Group 2	601.4 ± 66.9	α	414.0 ± 48.8	α	139.8 ± 5.6	α	53.6 ± 1.5	α	352.4 ± 4.5	α
	Group 3	1527.0 ± 70.5	*, #	983.0 ± 88.5	*, #	156.0 ± 4.9	*, #	44.6 ± 2.2	*, #	328.4 ± 2.2	*, #
	Group 4	203.8 ± 25.0	*, #, α	120.6 ± 30.1	*, #, α	135.2 ± 3.5	α	60.6 ± 1.8	*, #, α, γ	388.5 ± 8.9	*, #, α
	Group 5	331.4 ± 20.5	*, #, α	241.2 ± 14.4	*, #, α	139.5 ± 2.4	α	54.1 ± 1.3	α	372.8 ± 7.5	*, #, α
	Group 6	647.8 ± 44.3	α	425.2 ± 51.2	α	146.6 ± 4.6	α, γ	51.2 ± 1.9	α	354.4 ± 11.4	α
Termination 2 (day 20)	Group 1	620.0 ± 66.6	α	392.2 ± 35.0	α	140.9 ± 3.8	α	50.7 ± 2.7	α	351.1 ± 1.9	α
	Group 2	650.0 ± 51.2	α	408.8 ± 17.1	α	140.8 ± 6.3	α	54.1 ± 1.6	α	354.6 ± 7.6	α
	Group 3	1534.4 ±126.2	*, #	1019.4 ± 80.5	*, #	155.6 ± 4.4	*, #	42.5 ± 2.3	*, #	328.0 ± 14.0	*, #
	Group 4	195.6 ± 15.7	*, #, α	138.4 ± 19.3	*, #, α	133.5 ± 5.6	α	56.4 ± 1.3	α	380.0 ± 11.4	*, #, α
	Group 5	321.6 ± 37.3	*, #, α	264.4 ± 27.7	*, #, α	136.5 ± 2.8	α	54.5 ± 2.7	α	372.8 ± 6.9	*, #, α
	Group 6	566.2 ± 60.9	α	414.0 ± 48.6	α	138.9 ± 3.3	α	49.7 ± 4.6	-	343.0 ± 9.7	-
	Group 7	632.9 ± 72.8	α	422.8 ± 51.9	α	144.7 ± 3.6	α	48.4 ± 3.0	#, α	356.0 ± 10.0	α
	Group 8	974.2 ± 130.8	*, #, α	537.0 ± 67.0	*, #, α	149.0 ± 4.0	*, #	48.1 ± 2.0	#, α	346.3 ± 5.3	α
Termination 3 (day 28)	Group 1	652.4 ± 104.9	α	377.2 ± 20.8	α	141.6 ± 1.5	α	51.9 ± 2.5	α	350.9 ± 3.8	-
	Group 2	629.0 ± 48.1	α	394.0 ± 38.7	α	145.4 ± 2.2	α	53.8 ± 1.2	α	351.0 ± 4.7	-
	Group 3	1763.8 ± 78.2	*, #,γ	1052.4 ± 123.9	*, #	153.6 ± 3.0	*, #	42.2 ± 3.1	*, #	334.9 ± 17.3	-
	Group 4	209.6 ± 42.2	*, #, α	129.8 ± 9.5	*, #, α	135.1 ± 4.3	*, #,α	59.9 ± 2.6	*, α	380.9 ± 12.5	-
	Group 5	300.6 ± 23.5	*, #, α	249.2 ± 39.1	*, #, α	130.1 ± 1.2	#, α, γ	57.7 ± 3.2	α	373.8 ± 7.4	*, #
	Group 6	568.2 ± 104.8	α	416.0 ± 41.5	α	140.3 ± 2.5	α	54.4 ± 3.8	α	356.9 ± 6.4	-
	Group 7	557.0 ± 68.8	α	418.0 ± 69.8	α	142.1 ± 5.7	-	53.6 ± 3.9	α, γ	356.7 ± 7.9	-
	Group 8	928.2 ± 31.2	*, #, α	480.2 ± 24.4	*, α	153.1 ± 4.2	*	48.1 ± 3.5	-	332.3 ± 5.9	*, #, γ
Termination 4 (day 34)	Group 1	641.4 ± 39.5	α	410.2 ± 35.9	α	142.8 ± 3.4	α	52.5 ± 2.4	α	346.0 ± 4.8	α
	Group 2	651.4 ± 37.0	α	397.7 ± 52.5	α	142.1 ± 3.2	α	54.2 ± 2.4	α	352.1 ± 7.5	α
	Group 3	1607.6 ±150.8	*, #	1119.8 ± 90.7	*, #	155.7 ± 5.0	*, #	43.7 ± 2.2	*, #	323.2 ± 11.4	*, #
	Group 4	205.3 ± 21.0	*, #, α	126.3 ± 15.5	*, #, α	135.0 ± 5.5	*, #, α	62.1 ± 2.4	*, #, α, γ	381.3 ± 9.2	*, #, α
	Group 5	323.3 ± 48.5	*, #, α	223.7 ± 26.2	*, #, α	133.9 ± 4.5	*, #, α	57.4 ± 3.5	*, α	359.6 ± 14.5	α
	Group 6	589.8 ± 60.6	α	428.0 ± 36.8	α	141.8 ± 2.6	α	52.2 ± 3.4	α	354.4 ± 12.0	α
	Group 7	643.7 ± 61.7	α	409.2 ± 50.9	α	142.0 ± 3.4	α	50.0 ± 2.6	#, α	355.2 ± 8.4	α
	Group 8	967.5 ± 74.5	*, #, α	504.6 ± 57.6	*, #, α	150.4 ± 4.0	*, #, α	44.4 ± 2.5	*, #, γ	325.4 ± 10.3	*, #, α, γ

Results are presented as mean values ± SD. Statistical analyses were performed by the Kruskal–Wallis test and post hoc Dunn’s test or ANOVA and post hoc Tukey’s test. The significance of the differences between controls and particular treated groups are observed and marked by asterisks (*) vs. saline control; (#) vs. phage control (group 2); (α) vs. infected control (group 3); (γ) vs. termination 2.

## Data Availability

Not applicable.

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
