# Peer review of "Phage Therapy vs. the Use of Antibiotics in the Treatment of Salmonella-Infected Chickens: Comparison of Effects on Hematological Parameters and Selected Biochemical Markers"

_antibiotics, 2022, doi:10.3390/antibiotics11121787_

Round 1
Reviewer 1 Report
The Authors properly described the research methods and discussed the results of this study. I have comments on the Discussion section and Conclusion.
In my opinion, this article can add important data to enrich the knowledge in this area. However, numerous corrections are needed (please see below).
1. In line 30 - subsequent keywords should be written without bold
2. Please remove the space between lines 65 and 66
3. In line 93 - I recommend to change “count of erythrocyte number” to “erythrocyte count”
4. In line 95 - post hoc should be in italics
5. In line 111 - I recommend to change “count of platelets number” to “platelets count”
6. In line 113 - post hoc should be in italics
7. Figures 1, 2 and 3 - I recommend to delete the description above the figure (erythrocyte count / hematocrit / platelets) - the plot information is included in the description of Fig. 1, Fig. 2, and Fig. 3, respectively
8. Figures 1, 2 and 3 - I recommend to change the y-axis values, e.g. starting with higher values so that the differences can be seen more clearly
9. Figure 2 - y-axis - please make a space in [hematocrit (%)]
10. Figure 3 - units on the y-axis - should be (No x 106/μL)
11. Figures 1, 2 and 3 - y-axis - axis values should be dotted, for example 1.0 instead of 1.0
12. In line 142 - I recommend to change “no statistically significant differences” to “no significant differences”
13. In line 147 - I recommend to change “no statistically significant changes” to “no significant changes”
14. In line 157 - I recommend to change “Changes in the counts of absolute monocyte count” to “Changes in the absolute monocyte count”
15. In line 160 - please remove the double space between [saline control;] and [(#) vs…]
16. Table 1 - description of columns in the table - units - should be L instead of l (column 3/5/7/11)
17. In lines 167, 173, and 180 - Salmonella should be in italics
18. Figures 4, 5 and 6 - I recommend to delete the description above the figure (alanine transaminase (ALT) / aspartate aminotransferase (AST) / C-reactive protein (CRP)) - the plot information is included in the description of Fig. 4, Fig. 5, and Fig. 6, respectively
19. In line 186 - post hoc should be in italics
20. Figure 6 - units on the y-axis - should be (ng/mL)
21. Figures 4, 5 and 6 - I recommend to change the y-axis values, e.g. Fig. 4 - from 0 to 200; Fig. 5 - from 0 to 400; Fig. 6 - from 0 to 8000, so that the differences can be seen more clearly
22. In lines 217-218 - I recommend to change “S. enterica cells (serovars Typhimurium and Enteritidis)” to “S. enterica serovars Typhimurium and Enteritidis”
23. Please remove the space between lines 225 and 226; 235 and 236; 253 and 254
24. In line 266 - please remove the space from reference numbers - should be [15,16]. The same in line 283.
25. In line 274 - please make spaces between the temperature value and the unit (33 °C instead of 33°C; 22 °C instead of 22°C). The same in lines 296, 298, 300, 302, 305, 309, 310, 353, 354, 387, 389, 393, 395, and 397.
26. What strain number of the S. enterica serovar Heidelberg was used in the study?
27. In line 297 - should be mL instead of ml. The same in lines 304, 315, 326, 337, 345, 350, 404 (mL) and 356, 361-362, 388, 391, 394, 396, 398 (μL).
28. In line 410 - post hoc should be in italics
29. Author Contribution - this section should be edited in accordance with the journal's guidelines
30. Lack of Conclusion section in the manuscript.
31. The discussion session should be re-written. Much of this is more of a conclusion that is missing at the end of the manuscript.
32. References - should be corrected according to guidelines. When using abbreviations of authors' names, no spaces should be included; the names of microorganisms should be written in italics; references 14, 15, 16, 17 and 26 - there is no need to enter the letters a and b for the year of publication - it is not in accordance with the guidelines of this journal.
Author Response
Reviewer #1
- In line 30 - subsequent keywords should be written without bold
ANSWER: Thank you very much for bringing this to our attention.
Keywords: Phage therapy; Salmonella infection; chicken; antibiotics; phage cocktail
- Please remove the space between lines 65 and 66
ANSWER: Thank you for your suggestion, the space between lines 65 and 66 has been removed.
- In line 93 - I recommend to change “count of erythrocyte number” to “erythrocyte count”
ANSWER: Thank you for your comment, the sentence has been changed (lines: „103 – 104”).
Changes in the erythrocyte count number in the blood of chicken receiving phage cocktail or antibiotics after 6, 20, 28 and 34 days of experiments.
- In line 95 - post hoc should be in italics
ANSWER: Thank you for your suggestion, the sentence has been corrected (line: „105”).
Statistical analyses were performed by ANOVA and post-hoc Tukey test.
- In line 111 - I recommend to change “count of platelets number” to “platelets count”
ANSWER: Thank you for your comment, the sentence has been changed (lines: „141 – 142”).
Changes in the platelets count number in the blood of chicken receiving phage cocktail or antibiotics after 6, 20, 28 and 34 days of experiments.
- In line 113 - post hoc should be in italics
ANSWER: Thank you very much for bringing this to our attention. (line: „143”)
Statistical analyses were performed by ANOVA and post-hoc Tukey test.
- Figures 1, 2 and 3 - I recommend to delete the description above the figure (erythrocyte count / hematocrit / platelets) - the plot information is included in the description of Fig. 1, Fig. 2, and Fig. 3, respectively
ANSWER: Thank you very much for your comment, we have deleted the description above the figures.
- Figures 1, 2 and 3 - I recommend to change the y-axis values, e.g. starting with higher values so that the differences can be seen more clearly
ANSWER: Thank you very much for your suggestion, we have changed the y-axis values.
- Figure 2 - y-axis - please make a space in [hematocrit (%)]
ANSWER: Thank you very much for bringing this to our attention, we have changed the y-axis description in the Figure 2.
- Figure 3 - units on the y-axis - should be (No x 106/μL)
ANSWER: Thank you very much for your comment, we have changed the units on the y-axis in the Figure 3.
- Figures 1, 2 and 3 - y-axis - axis values should be dotted, for example 1.0 instead of 1.0
ANSWER: Thank you very much for your suggestion, the y-axis values have been corrected in the all mentioned figures.
- In line 142 - I recommend to change “no statistically significant differences” to “no significant differences”
ANSWER: Thank you for your comment, the sentence has been changed (lines: „172 – 174”).
Although no significant differences were observed in group 6 relative to the uninfected control groups, such differences were observed in groups 7 and 8 at termination 2 and termination 4.
- In line 147 - I recommend to change “no statistically significant changes” to “no significant changes”
ANSWER: Thank you for your suggestion, the sentence has been changed (lines: „177 – 178”).
No significant changes were observed in group 6 and group 7 compared to uninfected control groups throughout the experiment.
- In line 157 - I recommend to change “Changes in the counts of absolute monocyte count” to “Changes in the absolute monocyte count”
ANSWER: Thank you for your comment, the sentence has been changed (lines: „186 – 187”).
Changes in the absolute monocyte count, absolute eosinophil count, MCV (Mean Corpuscular Volume), MCH (Mean Corpuscular Hemoglobin), MCHC (Mean Corpuscular Hemoglobin Concentration) of chickens receiving phage cocktail or antibiotics after 6, 20, 28 and 34 days of experiments.
- In line 160 - please remove the double space between [saline control;] and [(#) vs…]
ANSWER: Thank you very much for bringing this to our attention, the double space has been removed.
- Table 1 - description of columns in the table - units - should be L instead of l (column 3/5/7/11)
ANSWER: As suggested, a description of columns in the table was changed.
- In lines 167, 173, and 180 - Salmonella should be in italics
ANSWER: Thank you very much for bringing this to our attention, the indicated sentences have been corrected lines: „219 – 220”; „223 – 227” and „231 – 232”).
The results show a significant increase in ALT levels in group 3, infected with Salmonella, but not treated throughout the experiment (Figure 4).
. In contrast, when the phage cocktail was used, as well as in the Salmonella-infected groups where it was applied 24 hours after infection (group 6) or after 2 days (group 7), the level of the enzyme tested remained low, within the normal range throughout the experiment.
With regard to CRP (Figure 6), its significantly elevated level was shown only in Salmonella-infected group 3 and in group 8, where phage cocktail was implemented latest.
- Figures 4, 5 and 6 - I recommend to delete the description above the figure (alanine transaminase (ALT) / aspartate aminotransferase (AST) / C-reactive protein (CRP)) - the plot information is included in the description of Fig. 4, Fig. 5, and Fig. 6, respectively
ANSWER: Thank you very much for your comment, we have deleted the description above the figures.
- In line 186 - post hoc should be in italics
ANSWER: Thank you very much for bringing this to our attention. (line: „251”)
Statistical analyses were performed by Kruskal-Wallis test and post-hoc Dunn test.
- Figure 6 - units on the y-axis - should be (ng/mL)
ANSWER: Thank you very much for your comment, we have changed the units on the y-axis in the Figure 6.
- Figures 4, 5 and 6 - I recommend to change the y-axis values, e.g. Fig. 4 - from 0 to 200; Fig. 5 - from 0 to 400; Fig. 6 - from 0 to 8000, so that the differences can be seen more clearly
ANSWER: Thank you very much for your suggestion, we have changed the y-axis values in all mentioned figure.
- In lines 217-218 - I recommend to change “S. entericacells (serovars Typhimurium and Enteritidis)” to “S. enterica serovars Typhimurium and Enteritidis”
ANSWER: Thank you very much for your suggestion, the sentence has been changed (lines: „301 - 304”)
Our previous studies led to development of a phage cocktail, composed of bacteriophages vB_SenM-2 and vB_Sen-TO17, which is effective in eliminating S. enterica serovars Typhimurium and Enteritidis and safe to the G. mellonella animal model as well as in cell culture tests.
- Please remove the space between lines 225 and 226; 235 and 236; 253 and 254
ANSWER: Thank you very much for bringing this to our attention, all mentioned space have been removed.
- In line 266 - please remove the space from reference numbers - should be [15,16]. The same in line 283.
ANSWER: Thank you very much for bringing this to our attention, all mentioned spaces have been removed.
- In line 274 - please make spaces between the temperature value and the unit (33 °C instead of 33°C; 22 °C instead of 22°C). The same in lines 296, 298, 300, 302, 305, 309, 310, 353, 354, 387, 389, 393, 395, and 397.
ANSWER: Thank you very much for your comment, all required spaces have been added.
- What strain number of the S. entericaserovar Heidelberg was used in the study?
ANSWER: The strain used is from the collection of the Department of Molecular Biology, Faculty of Biology, University of Gdansk and has no number assigned.
- In line 297 - should be mL instead of ml. The same in lines 304, 315, 326, 337, 345, 350, 404 (mL) and 356, 361-362, 388, 391, 394, 396, 398 (μL).
ANSWER: Thank you very much for bringing this to our attention, all units have been corrected.
- In line 410 - post hoc should be in italics
ANSWER: Thank you very much for bringing this to our attention, the indicated sentences has been corrected (lines: „552 – 554”).
If the assumptions of normality of distribution and/or homogeneity of variance were not met, the Kruskal–Wallis test and post-hoc Dunn test were applied.
- Author Contribution - this section should be edited in accordance with the journal's guidelines
ANSWER: We are very grateful for this comment, author contribution section has been edited (lines: „558 – 563”).
Author Contributions: Conceptualization, Ł.G. and M.P.; methodology, Ł.G. and M.P.; validation, Ł.G. and M.P.; formal analysis, Ł.G., A.W., G.W. and M.P.; investigation, Ł.G., A.W., G.W. and M.P.; writing—original draft preparation, Ł.G., A.W., G.W. and M.P.; writing—review and editing, Ł.G., A.W., G.W., and M.P.; visualization, Ł.G. and M.P.; supervision, A.W. and G.W.; project administration, A.W.; funding acquisition, A.W. All authors have read and agreed to the published version of the manuscript.
- Lack of Conclusion section in the manuscript.
ANSWER: We are very grateful for this comment, conclusion section has been added (lines: „392 – 401”).
Conclusions
Our results indicate significantly less adverse effects of the phage cocktail relative to tested antibiotics. The latter agents caused deleterious changes in red blood cells parameters, including hematocrit, MCV, MCH, and MCHC. On the contrary, Salmonella-induced changes in the aforementioned parameters were normalized by the use of the phage cocktail. Furthermore, bacteriophages administered either immediately or two days after infection did not significantly affect the number of lymphocytes, monocytes and neutrophils in the serum. Administration of the tested antibiotics also caused increased activities of ALT and AST, suggesting the hepatotoxicity of these compounds. This was again in contrast to the phage cocktail which did not influence the activities of these enzymes.
- The discussion session should be re-written. Much of this is more of a conclusion that is missing at the end of the manuscript.
ANSWER: Thank you for your suggestion, the discussion section has been re-written (lines: „293 – 394”).
Phage therapy is considered as one of the possible alternatives to the use of antibiotics to combat pathogenic bacteria [18]. Although introduction of this method as an officially approved therapy to treat patients still requires both intensive studies and changes in current law, it appears that the use of bacteriophages in treatment of infected animals might be possible significantly earlier, especially due to differences in regulations between medical and veterinary rules [19]. Nevertheless, before formal recommendations to use phage therapy to treat animals can be issued, both efficacy and safety of this method should be confirmed.
Our previous studies led to development of a phage cocktail, composed of bacteriophages vB_SenM-2 and vB_Sen-TO17, which is effective in eliminating S. enterica serovars Typhimurium and Enteritidis and safe to the G. mellonella animal model as well as in cell culture tests [14]. Moreover, this cocktail was as effective in treatment of S. enterica-infected chickens as antibiotics (enrofloxacin or colistin) while causing significantly less adverse effects than those drugs as estimated by changes in gut microbiome [15] or disturbance of the immune response [16]. On the other hand, effects of these two methods of treatment were not compared previously in the light of changes in hematological parameters and biochemical markers. Thus, this work was conducted to fill this gap in our knowledge.
Hematocrit is an indicator that determines the volume ratio of erythrocytes to whole blood. A decrease in this index, as well as an insufficient erythrocyte count, can indicate a number of abnormalities, such as gastrointestinal bleeding, bone marrow disturbances, kidney damage or, most commonly, the onset of anemia, resulting from iron deficiency. Iron determines the supply of adequate amounts of oxygen, electron transport, and the proper functioning of enzymes. It is important for high-metabolic-rate cells [17]. Hematocrit is of particular importance in the assessment of animal physiology. Some mammals, such as dogs or horses, can store erythrocytes in the spleen and thus modulate hematocrit or hemoglobin levels depending on the intensity of exercise. Birds, on the other hand, which do not have such a regulatory mechanism, are able to lower their hematocrit values by hemodilution in response to intensive exercise or exposure to aversive factors [18]. When considering the effect of administered substances on erythrocyte indices, including the hematocrit value, physiological factors such as age, gender and hormone levels, should also be taken into account [19]. Other indicators also play an important role when it comes to assessing the organism’s overall condition. The MCV provides information on the average volume of a single erythrocyte and allows early detection of anemia. Its value depends on plasma osmolarity and the rate of erythrocyte division. The MCH is an indicator of the average mass of hemoglobin in a single red blood cell. Together with MCHC value, it is useful in distinguishing between different types of anemia [19]. Changes in hematological parameters are useful for assessing the organism’s adaptation to adverse conditions or stressors. This is particularly important in the case of industrial poultry rearing for meat and eggs. Erythrocyte indices change significantly depending on husbandry conditions and nutritional status. Overly nutrient-poor feed and prolonged exposure of poultry to stress factors results in lower erythrocyte counts and hemoglobin levels, leading to erythrocytopenia and reduced organism performance. An increase in ambient temperature, result in the loss of a large amount of liquid through the respiratory system, which then leads to a decrease in plasma volume and an increase in hematocrit level. Similarly, in the case of dehydration by evaporation process, there is a significant increase in hematocrit values. In contrast, a non-physiological reduction in hematocrit levels through the hemodilution process occurs with severe stressors, especially heat stress exposure [20].
As indicated by results shown in this report, antibiotic therapy, which in poultry is sometimes administered prophylactically, also for chicken growth promotion or therapeutically [21], can be a type of negative stressor that causes a number of disturbances in the organism, especially if we consider the immune system [16] and in particular, erythrocyte indices. This is important because it has a number of negative consequences, including economic ones, due to the deterioration of the quality of the meat, the lower number of eggs laid or the welfare of consumers (adverse effects on the intestinal microbiome and the development of drug-resistant bacteria) [21]. The search for alternative methods for controlling bacterial infections is therefore the issue that needs to be addressed intensively, especially when it comes to infections with Salmonella strains, which pose serious challenges to the immune system of animals destined for consumption [22]. It was demonstrated that Salmonella enterica serotype Enteritidis leads in chickens to slowed growth of key immune organs, changes in the profile of important immune cell types, reduced antibody production, increased levels of pro-inflammatory cytokines and excessive activation of the stress axis, with consequent effects on hematological parameters as well [23]. This is also confirmed by our observations, which showed, among other things, a non-physiological increase in erythrocyte count, hematocrit level and MCV value. This type of deviation from reference values is most often observed in situation of severe dehydration resulting in blood thickening. Although there are insufficient reports in the literature of negative effects of antibiotic therapy on hematological parameters in poultry, there are papers available indicating a number of different side effects of administration of such popular veterinary antibiotics as enrofloxacin [24]. Only a few previously published reports addressed the problem investigated in this work. Nevertheless, it appears that results of all these studies are quite similar. A comparison of the efficacy of phages and antibiotics against acute pneumonia in a mouse model was described previously [27]. In that work, two phages (536_P1 and LM33_P1) and three antibiotics (ceftriaxone, cefoxitin, and imipenem – cilastatin) were used. The phages significantly and rapidly reduced the number of bacterial cells, and restored normal blood counts, otherwise disrupted by the development of an abrupt bacterial infection [28]. It should also be emphasized that rapid lysis of bacterial cells does not induce an increase in inflammatory markers as compared to the antibiotic therapy [29]. Our studies indicated that there is no significant change in the number of immunocompetent cells (lymphocytes, monocytes or neutrophils) in the serum after administration of bacteriophages vB_SenM-2 and vB_Sen-TO17. Analogous results were reported previously by others with respect to changes in the number of macrophages, T and B lymphocytes, and dendritic cells in peripheral organs (spleen, liver, and lymph nodes) after administration of other phages [30-32]. Our results also confirm that after the phage cocktail administration, the hematological parameters analyzed are not significantly different from the results obtained in the control groups. The normalization of erythrocyte counts following the phage cocktail has tangible physiological benefits, as it avoids disturbances in hematological parameters induced by bacterial infection, which lead to changes and have a negative impact on many organs, particularly the liver and spleen. However, it should be emphasized that only administration immediately or up to two days after detection of the presence of bacteria in the feces guarantees the effective action of the phage cocktail. When it is administered after a longer period, the bacterial infection is already so developed that it causes a number of negative changes that disrupt the functioning of many systems and organs, including, but not limited to, the spleen and liver, resulting in hepatosplenomegaly [33].
In summary, our results confirmed a favorable safety profile of the use of phage therapy (particularly the phage cocktail consisting of vB_SenM-2 and vB_Sen-TO17) in Salmonella-infected chickens. Moreover, this study indicated serious adverse effects of enrofloxacin and colistin on hematological parameters and ALT and AST activities in these birds. Therefore, the phage therapy with bacteriophages vB_SenM-2 and vB_Sen-TO17 may be further considered as an alternative method to either treat of or prevent infections of chickens with S. enterica serovar Typhimurium.
- References - should be corrected according to guidelines. When using abbreviations of authors' names, no spaces should be included; the names of microorganisms should be written in italics; references 14, 15, 16, 17 and 26 - there is no need to enter the letters a and b for the year of publication - it is not in accordance with the guidelines of this journal.
ANSWER: Thank you very much for bringing this to our attention, the references have been corrected according to the guidelines.

Reviewer 2 Report
Line 44: Change “filed” to “field”
Line 158 – 160: The text beginning from “Results are ………” should be transferred as a footnote beneath/under the table. Only the table title should remain
The discussion can be improved further by critically discussing some few results the authors were silent about. Some suggestions below to further improve on the discussion
- What is the implication of the significant decrease in the hematocrit and erythrocyte counts?
-what are the biological significance of the normalization of red blood cells in the phage-treated group at termination 2-4? Are they any benefits or downsides to this observation?
-The erythrocyte count and hematocrit as well as the MCV were elevated throughout the experiment for group 3. Are they are benefits of Salmonella infection in this respect or what is happening in this case/potential application?
-Are they any pathological implications for the decrease in the MCH and MCHC for group 3 and how it relates to the overall findings with respect to the phage treatment groups.
-You did discuss the implication of antibiotic treatment on the distortion of the erythrocytic indices but I will also suggest that the authors go further to discuss their findings with other body of literature and provide references to support or dispel their observation.
-In view of the observations after the experimental works following treatment with phage cocktail, of the 3 different treatment regimen for group 6, 7 and 8, could they authors possibly suggest which of the treatment approach will be best and possibly adopted viz-a-viz the biochemical and hematological findings.
Author Response
Reviewer #2
Line 44: Change “filed” to “field”
ANSWER: Thank you very much for bringing this to our attention, the sentence has been re-written.
Line 158 – 160: The text beginning from “Results are ………” should be transferred as a footnote beneath/under the table. Only the table title should remain
ANSWER: Thank you very much for your suggestion, the table description has been corrected.
The discussion can be improved further by critically discussing some few results the authors were silent about. Some suggestions below to further improve on the discussion
- What is the implication of the significant decrease in the hematocrit and erythrocyte counts?
ANSWER: Thank you very much for your comment, the mentioned implication has been added to the discussion section (lines: „311 – 324”).
Hematocrit is an indicator that determines the volume ratio of erythrocytes to whole blood. A decrease in this index, as well as an insufficient erythrocyte count, can indicate a number of abnormalities, such as gastrointestinal bleeding, bone marrow disturbances, kidney damage or, most commonly, the onset of anemia, resulting from iron deficiency. Iron determines the supply of adequate amounts of oxygen, electron transport and the proper functioning of enzymes. It is important for high-metabolic-rate cells [17]. Hematocrit is of particular importance in the assessment of animal physiology. Some mammals, such as dogs or horses, can store erythrocytes in the spleen and thus modulate hematocrit or hemoglobin levels depending on the intensity of exercise. Birds, on the other hand, which do not have such regulatory mechanism, are able to lower their hematocrit values by hemodilution in response to intensive exercise or exposure to aversive factors [18]. When considering the effect of administered substances on erythrocyte indices, including the hematocrit value, physiological factors such as age, gender and hormone levels, should also be taken into account [19].
-what are the biological significance of the normalization of red blood cells in the phage-treated group at termination 2-4? Are they any benefits or downsides to this observation?
ANSWER: Thank you for your comment. The normalization of erythrocyte counts following the phage cocktail has tangible physiological benefits, as it avoids disturbances in hematological parameters induced by bacterial infection, which lead to changes and have a negative impact on many organs, particularly the liver and spleen.
-The erythrocyte count and hematocrit as well as the MCV were elevated throughout the experiment for group 3. Are they are benefits of Salmonella infection in this respect or what is happening in this case/potential application?
ANSWER: Thank you very much for your comment, the mentioned implication has been added to the discussion section (lines: „348 – 356”).
The search for alternative methods for controlling bacterial infections is therefore the issue that needs to be addressed intensively, especially when it comes to infections with Salmonella strains, which pose serious challenges to the immune system of animals destined for consumption [22]. It was demonstrated that Salmonella enterica serotype Enteritidis leads in chickens to slowed growth of key immune organs, changes in the profile of important immune cell types, reduced antibody production, increased levels of pro-inflammatory cytokines and excessive activation of the stress axis, with consequent effects on hematological parameters as well [23].
-Are they any pathological implications for the decrease in the MCH and MCHC for group 3 and how it relates to the overall findings with respect to the phage treatment groups.
ANSWER: Thank you very much for your comment, the mentioned implication has been added to the discussion section (lines: „327 – 348”).
The MCH is an indicator of the average mass of hemoglobin in a single red blood cell. Together with MCHC value it is useful in distinguishing between different types of anemia [19]. Changes in hematological parameters are useful for assessing the organism’s adaptation to adverse conditions or stressors. This is particularly important in the case of industrial poultry rearing for meat and eggs. Erythrocyte indices change significantly depending on husbandry conditions and nutritional status. Overly nutrient-poor feed and prolonged exposure of poultry to stress factors results in lower erythrocyte counts and hemoglobin levels, leading to erythrocytopenia and reduced organism performance. An increase in ambient temperature, result in the loss of a large amount of liquid through the respiratory system, which then leads to a decrease in plasma volume and an increase in hematocrit level. Similarly, in the case of dehydration by evaporation process, there is a significant increase in hematocrit values. In contrast, a non-physiological reduction in hematocrit levels through the hemodilution process occurs with severe stressors, especially heat stress exposure [20]. As indicated by results shown in this report, antibiotic therapy, which in poultry is sometimes administered prophylactically, also for chicken growth promotion or therapeutically [21], can be a type of negative stressor that causes a number of disturbances in the organism, especially if we consider the immune system [16] and in particular, erythrocyte indices. This is important because it has a number of negative consequences, including economic ones, due to the deterioration of the quality of the meat, the lower number of eggs laid or the welfare of consumers (adverse effects on the intestinal microbiome and the development of drug-resistant bacteria) [21].
-You did discuss the implication of antibiotic treatment on the distortion of the erythrocytic indices but I will also suggest that the authors go further to discuss their findings with other body of literature and provide references to support or dispel their observation.
ANSWER: Thank you for your comment, the discussion section has been re-written (lines: „362 – 376”).
Only a few previously published reports addressed the problem investigated in this work. Nevertheless, it appears that results of all these studies are quite similar. A comparison of the efficacy of phages and antibiotics against acute pneumonia in a mouse model was described previously [27]. In that work, two phages (536_P1 and LM33_P1) and three antibiotics (ceftriaxone, cefoxitin, and imipenem – cilastatin) were used. The phages significantly and rapidly reduced the number of bacterial cells, and restored normal blood counts, otherwise disrupted by the development of an abrupt bacterial infection [28]. It should also be emphasized that rapid lysis of bacterial cells does not induce an increase in inflammatory markers as compared to the antibiotic therapy [29]. Our studies indicated that there is no significant change in the number of immunocompetent cells (lymphocytes, monocytes or neutrophils) in the serum after administration of bacteriophages vB_SenM-2 and vB_Sen-TO17. Analogous results were reported previously by others with respect to changes in the number of macrophages, T and B lymphocytes, and dendritic cells in peripheral organs (spleen, liver, and lymph nodes) after administration of other phages [30-32].
-In view of the observations after the experimental works following treatment with phage cocktail, of the 3 different treatment regimen for group 6, 7 and 8, could they authors possibly suggest which of the treatment approach will be best and possibly adopted viz-a-viz the biochemical and hematological findings.
ANSWER: Thank you for your comment, the discussion section has been re-written (lines: „376 – 387”).
Our results also confirm that after the phage cocktail administration, the hematological parameters analyzed are not significantly different from the results obtained in the control groups. The normalization of erythrocyte counts following the phage cocktail has tangible physiological benefits, as it avoids disturbances in hematological parameters induced by bacterial infection, which lead to changes and have a negative impact on many organs, particularly the liver and spleen. However, it should be emphasized that only administration immediately or up to two days after detection of the presence of bacteria in the feces guarantees the effective action of the phage cocktail. When it is administered after a longer period, the bacterial infection is already so developed that it causes a number of negative changes that disrupt the functioning of many systems and organs, including, but not limited to, the spleen and liver, resulting in hepatosplenomegaly [33].

Reviewer 3 Report
The manuscript entitled ‘Phage therapy versus the use of antibiotics in the treatment of Salmonella-infected chickens: comparison of effects on hematological parameters and selected biochemical markers’ collects useful information on the use of alternatives to antibiotics against Salmonella in chickens. The use of bacteriophages could be applied in the context of ‘one health’ to reduce the risk associated to the abuse of antiobiotics in farm animals and stablishes the basis to apply this alternative in other areas. The manuscript is well organised, and the results are clear. Only some aspects are recommended to improve the article.
Abstract: Remove ‘induction of’ because is redundant to write induction of hepatotoxicity was suggested to be induced…
Line 64: Remove the word especially
Figure 1: It should be more clear which termination corresponds with each day. The sentence ‘Statistical analyses were performed by ANOVA and post-hoc Tukey test.’ can be removed. I would add information about the treatments from each group in the figure caption. These recommendations can be extrapolated to the rest of the figures and Table 1.
Figure 3: In termination 2 the group 4 seems to have differences with Groups 1 and 2. Please, check this data.
Table 1: The value for p in MCV of Group 7 from termination 3 is in blank.
Discussion: Add information about why the latest administration of the combination of the phages increased the level of enzimes related with hepatic damage (group 8). It seems that rapid detection and application of phages is necessary to avoid liver damage.
References: Write Samonella througout the references
Lines 468 and 469: in vitro, Galleria mellonella
Line 508: Marrubium vulgare
Author Response
Reviewer #3:
Abstract: Remove ‘induction of’ because is redundant to write induction of hepatotoxicity was suggested to be induced…
ANSWER: Thank you for your comment, the abstract section has been re-written.
Line 64: Remove the word especially
ANSWER: Thank you very much for bringing this to our attention, the word especially has been removed.
Figure 1: It should be more clear which termination corresponds with each day. The sentence ‘Statistical analyses were performed by ANOVA and post-hoc Tukey test.’ can be removed. I would add information about the treatments from each group in the figure caption. These recommendations can be extrapolated to the rest of the figures and Table 1.
ANSWER: Thank you very much for your comment, the number of days is indicated for each termination. Thank you for your suggestion, however, we cannot remove the information on what type of statistical test was used, as the statistical analysis varied for the different parameters. In addition, including information on the figures as to what was given in a particular group will limit their readability (for the sake of example, we include one figure, modified as suggested).
Figure 3: In termination 2 the group 4 seems to have differences with Groups 1 and 2. Please, check this data.
ANSWER: Thank you for your suggestion, the differences mentioned were below the threshold for statistical significance, the p-value was 0.051 (Group 4 vs. Group 1) and 0.052 (Group 4 vs. Group 2), respectively.
Table 1: The value for p in MCV of Group 7 from termination 3 is in blank.
ANSWER: Thank you very much for bringing this to our attention, the values in the table 1 have been checked and corrected.
Discussion: Add information about why the latest administration of the combination of the phages increased the level of enzimes related with hepatic damage (group 8). It seems that rapid detection and application of phages is necessary to avoid liver damage.
ANSWER: We are grateful for this comment, the discussion section has been re-written (lines: „376 – 387”).
Our results also confirm that after the phage cocktail administration, the hematological parameters analyzed are not significantly different from the results obtained in the control groups. The normalization of erythrocyte counts following the phage cocktail has tangible physiological benefits, as it avoids disturbances in hematological parameters induced by bacterial infection, which lead to changes and have a negative impact on many organs, particularly the liver and spleen. However, it should be emphasized that only administration immediately or up to two days after detection of the presence of bacteria in the feces guarantees the effective action of the phage cocktail. When it is administered after a longer period, the bacterial infection is already so developed that it causes a number of negative changes that disrupt the functioning of many systems and organs, including, but not limited to, the spleen and liver, resulting in hepatosplenomegaly [33].
References: Write Samonella througout the references
Lines 468 and 469: in vitro, Galleria mellonella
Line 508: Marrubium vulgare
ANSWER: Thank you very much for bringing this to our attention, all mentioned references have been corrected (lines: „605 – 607”; „662 – 663”).

Reviewer 4 Report
Authors do not include a methods section, including a description of the experimental design of the chicken experiments. Can't continue review without it.
Extensive editing of grammar is required, and I include some edits suggested for the introduction in a separate file. Authors should use a professional editing service to improve presentation.

Author Response
Reviewer #4
Authors do not include a methods section, including a description of the experimental design of the chicken experiments. Can't continue review without it.
ANSWER: Contrary to the comment of the reviewer, the Methods section has been included in the first version of the manuscript, sent out for review. However, according to the style of this journal and to Instructions to Authors, the Methods section should be placed after Results and Discussion sections. Therefore, we have followed these instructions. Obviously, the Methods section is included in the revised manuscript (as section no. 4) and reads as follows:
- Materials and Methods
4.1. Animals
Details of the animals and experimental conditions were described previously [15,16]. Briefly, the experiment was conducted on non-genetically modified chickens (Gallus gallus domesticus) that were purchased from a licensed breeder (registration number PL28036602, Poland). The Experimental Infection Pavilon at the Department of Bird Diseases (Faculty of Veterinary Medicine, University of Warmia and Mazury, Olsztyn, Poland) was divided into 8 m2 boxes that held 25 chickens each. Living conditions were strictly controlled and monitored, with an average humidity of 75% under conditions of regular light-dark cycles (12-h day/12-h night, at light intensity 10 lx) and forced ventilation with 17 air changes per hour). Temperature was reduced from 33 °C (beginning of the experiment) to 22 °C (the end of the experiment). The chickens had unlimited access to forage and water. To reduce the risk of contamination, the entire research complex was equipped with a high-efficiency particulate absorbing (HEPA) filter system and automation to maintain a cascade of pressures in the sanitary corridors, boxes, and locks. All experiments were approved by the Local Ethics Committee for Experimental Animals in Olsztyn (permission no. 62/2019, dated on 30 July, 2019).
4.2. Bacteriophages and bacterial strain
The bacteriophages vB_Sen-TO17 and vB_SenM-2, used for the phage cocktail, were characterized previously [34,35] and their safety was confirmed in in vitro experiments with the chicken fibroblast model (UMNSAH/DF-1) and in in vivo studies with the Galleria mellonella animal model [14]. Salmonella enterica serovar Typhimurium (strain KOS 13) was obtained from the National Salmonella Center at the Medical University of Gdansk (Poland), and Salmonella enterica serovar Heidelberg came from the Collection of the Department of Molecular Biology, University of Gdansk. Isolation of S. Typhimurium in chicken fecal samples and cloacal swabs was conducted in accordance with ISO 65791:2017 standards and the previously described procedure [16].
4.3. The preparation of phage cocktail
Preparation of the phage lysates included in the cocktail used in the experiment was carried out in accordance with the previously published protocols [15]. In brief, an overnight culture of S. enterica (S. Heidelberg for phage vB_SenM-2 and S. Typhimurium for vB_SenTO17) after inoculation into fresh LB medium (BioShop, Burlington, Canada) at the ratio (v/v) of 1:100, and incubation at 37 °C with shaking (150 rpm) until OD600=0.15 (1.5 x 108 CFU/mL), was infected with the appropriate bacteriophage at a multiplicity of infection (m.o.i) of 0.5, and incubated at 37 °C until lysis. To purify the phage lysate, polyethylene glycol 8000 (PEG8000) (BioShop, Burlington, Canada) was added to final concentration of 10% and the lysate was then incubated with shaking overnight at 4 °C, using a magnetic stirrer (Carl Roth, Karlsruhe, Germany). Then, the lysate was centrifuged at 10,000 × g for 30 min at 4 °C (Avanti JXN-26, rotor JLA-8.100, Beckman Coulter, Indianapolis, IN, USA) and the obtained precipitate was suspended in 0.89% NaCl (Alchem, Torun, Poland). To remove PEG8000 completely, 2 mL of chloroform was added to the lysate and which was subjected to double centrifugation at 4,000 × g for 15 min at 4 °C (Avanti JXN-26, rotor JS-13.1, Beckman Coulter, Indianapolis, IN, USA). In the next step, the lysate was subjected to ultracentrifugation in a sucrose gradient (Sigma Aldrich, Saint Louis, MO, USA) at 95,000 × g (Optima XPN-100, rotor SW32.1 Ti, Beckman Coulter, Indianapolis, IN, USA) for 2.5 h at 10 °C. To remove residual sucrose, the lysate was dialyzed against 0.89% NaCl for 7 days at 4 °C. In order to exclude a possibility of contamination with bacterial endotoxin, the Purified Thermo ScientificTM LAL Chromogenic Endotoxin Quantitation Kit (Catalog No.: 12117850, Thermo Fisher Scientific Inc., Paisley, UK) was used. The obtained lysates were used to prepare a phage cocktail which was administered to the chickens. The purified and checked lysates of bacteriophages vB_SenM-2 and vB_Sen-TO17 were mixed in the 1:1 ratio (1 × 109 PFU/mL of each phage). Finally, the cocktail was suspended in 20 mM of CaCO3.
4.4. Animal groups and the schedule of the treatment
The detailed course of the experiment was described in two previously published papers [15, 16]. In brief, two hundred seven-day-old chickens were randomly divided into eight experimental groups, Group 1, receiving saline, and group 2, receiving phage cocktail from day 1 to day 15, were controls and were not infected with bacteria. For the former group, the aim was to see if the administration procedure could have a significant effect on the parameters studied, while the latter group was used to test a potential impact of the phage cocktail. Group 3 was the positive control, there were 25 Salmonella-infected chickens receiving saline until day 15 of the experiment. At day 0 of the experiment, groups 3-8 were infected by administering 1 mL of S. Typhimurium (106 CFU/mL) suspended in 0.89% NaCl into the beak. Twenty-four hours after infection (day 1 of the experiment), the chickens in group 4 started receiving enrofloxacin (Scanflox, Scanvet, Warsaw, Poland; dose 10 mg/kg per day), while 25 animals in group 5 were given colistin (Colisol, Ceva Animal Health, Warsaw, Poland; dose 120.000 IU/kg per day). For both groups, administration was continued for 5 days. Groups 6, 7, and 8 received the phage cocktail for 14 days. In the case of group 6, its administration began analogously to the antibiotics, 24 hours after infection, while animals in groups 7 and 8 started receiving the phage cocktail two and four days after confirmation of the bacteria in feces, respectively. The blood samples were taken at four time points while some of the animals from each group were sacrificed in a CO2 chamber. At day 6 of the experiment, after the end of antibiotic treatment, 5 mL of blood was collected from five chickens of each group (termination 1). At day 20 of the experiment, following the completion of phage therapy in group 6, blood was collected from another 5 chickens (in groups 1-6) and 10 chickens (in groups 7 and 8; termination 2). Subsequent blood sampling and sacrifice were performed at day 28 of the experiment (5 chickens from each group; termination 3) and day 34 of the experiment (10 chickens from each group; termination 4).
4.5. Blood collection
The blood sampling methodology was consistent with the previously published protocol [16]. Briefly, blood samples of 5 mL were collected from each chicken. To prevent blood clotting, the heparinized syringes tipped with a 25-gauge, 1-in-long needle and tubes containing sodium heparin were used. During the procedure, the animals were gently immobilized by holding and the needle was inserted into the brachial wing vein at a shallow angle (approximately 10-20°). Each blood sample was immediately divided according to the course of further determination: 1 mL of whole blood was used for morphological analyses (monocytes, eosinophils and red blood cells parameters) and flow cytometry (results of flow cytometric analyses were already published by Grabowski et al. [16], while the remaining blood was centrifuged (1,800 × g for 15 min at 4 °C) to obtain plasma, which was subjected to deep freezing (- 80 °C) until further analysis.
4.6 Analysis of selected blood morphological parameters
The morphological analysis of the collected whole blood sample (200 µL) was performed in the Horiba ABX Micros ES 60 automatic analyzer (Horiba Medical. Japan). Linearity specifications were determined by analyzing dilutions of a commercially available linearity control material that contains no interfering substances. In order to avoid meaningless results due to incorrect counts, the linearity range used for the particular parameters was: 0 – 99.9 K/µL for WBC; 0 – 8 M/µL for RBC; 0 – 24 g/dL for HGB; 50 – 200 fL for MCV; 0 – 2000 K/µL for PLT; 5 – 18 fL for MPV and 0 – 30% for RETIC %. The automatic analyzer used is commonly applied for the evaluation of hematological parameters in various animals species [36-38]. Following parameters were monitored: absolute number of monocytes and eosinophils, as well as red blood cell system indexes: erythrocyte count; hematocrit (HCT) level; mean red cell volume (MCV); mean corpuscular hemoglobin concentration (MCHC), and platelet number.
4.7. Determination of alanine transaminase (ALT) and aspartate aminotransferase (AST) concentrations in peripheral blood plasma
Levels of blood biochemical parameters, such as alanine transaminase (ALT) and aspartate aminotransferase (AST), were determined using an automated Architect c8000 Abbott biochemical analyzer (Abbott, Chicago, Illinois, United States). It is a fully-automated system that performs sample processing using potentiometric and photometric methods. The relevant calibration parameters and chicken-dedicated reference values were configured into the system and validated prior to the main analysis. For the determination of the above parameters, the following reference interval was used: 19–21 (IU/L) for ALT and 131–486 (IU/L) for AST, respectively. It was defined on the basis of Merck Veterinary Manual (2011).
4.8. Determination of c-reactive protein (CRP) concentration (using ELISA) in peripheral blood plasma
The measurement of the CRP level was carried out using an ELISA immunoenzymatic assay, based on the formation of bonds between antigen and antibody which are revealed by the color reaction with immunoglobulin-conjugated enzymes and their respective substrates. The procedure was performed according to the manual, included in the set of commercially available reagents (Catalog No.: ELK2038, ELK Biotechnology CO. ltd, Wuhan, China) and the method described previously [16]. All reagents and samples were brought to room temperature (20-25 °C) before use. Into each well of the titration plate (96-well Nunc plate), coated with CRP-specific monoclonal antibodies, 100 µL of buffer, test samples or standards were added in duplicate. Plate was covered and incubated at
37 °C for 80 min. Then, the plate contents were drained and washed with prepared buffer three times to remove an excess of unbound antigens. Next, 100 µL of a solution of specific polyclonal biotinylated antibody, conjugated with the enzyme for CRP, were added. The plate was covered, incubated at 37 °C for 50 min, then drained again, and washed three times. Subsequently, 100 µL of streptavidin-labeled horseradish peroxidase enzyme solution was added and incubated for 50 min at 37 °C. The plate was then drained and washed five times, and 90 µL of 3,3’5,5’-tetramethylbenzidine (a colored substrate for horseradish peroxidase) solution was added, and incubated for 20 min in the dark at 37 °C. The reaction was stopped by adding 50 µL of the blocking solution which changed the color of the product (from blue to yellow). Absorbance was measured at 10 min after stopping the reaction, using a Multiskan FC microplate reader (Thermo Fisher Scientific, Waltham, MA, USA), coupled with Skanlt 6.1.1. RE software which analyzes spectrophotometric color intensity, plots a standard curve based on the standards used, and reads the concentration values of the CRP in the plasma samples tested. The results obtained were given in ng/mL. The minimum sensitivity for the test was 0.32–20 ng/mL.
4.9. Statistical analysis
The results are presented as mean ± standard deviation (SD). For statistical analysis of the results, SPSS 21.0 (SPSS Inc., Armonk, NY, USA) software was used. The normality of the distribution of variables was checked with the Kolmogorov–Smirnov test and the homogeneity of the variances with Levene’s test. If the assumptions of normality of distribution and/or homogeneity of variance were not met, the Kruskal–Wallis test and post-hoc Dunn test were applied. Once both assumptions were met, the analysis was carried out on the basis of ANOVA and post-hoc Tukey’s test. Statistically significant differences were considered when p<0.05.
Extensive editing of grammar is required, and I include some edits suggested for the introduction in a separate file. Authors should use a professional editing service to improve presentation.
ANSWER: The introduction has been revised as suggested in a separate file, in addition the manuscript has been sent for professional editing service. However, due to administrative difficulties (payment for the service by the university), the language proofreading commissioned to the MDPI publisher did not make it in time to respond to the reviewers' comments.

Round 2
Reviewer 4 Report
LN 63. Indent to start paragraph
LN 72-73. Confusing sentence. Suggest “Whether bacteriophages may affect blood morphotic elements was examined and a comprehensive analysis of these phenomena was completed.
LN 315-392. Should be broken up into 4 paragraphs
LN315-328
LN328-345 Start at “Other indicators….”
LN345-366 Start at “As indicated by….”
LN366-392 Start at “Only a few….”
Author Response
We are grateful for all comments. All corrections have been incorporated into the attached version of the manuscript.
